# Large ensemble of downscaled historical daily snowfall from an Earth System Model to 5.5km resolution over Dronning Maud Land, Antarctica

Nicolas Ghilain[1], Stéphane Vannitsem[1], Quentin Dalaiden[2], Hugues Goosse[2], Lesley De Cruz[1,3], and Wenguang Wei[4]

[1]Royal Meteorological Institute of Belgium, Brussels, Belgium
[2]Georges Lemaître Centre for Earth and Climate Research, Earth and Life Institute, Université catholique de Louvain, Louvain-la-Neuve, Belgium
[3]Department of Electronics and Informatics, Vrije Universiteit Brussel, Brussels, Belgium
[4]Key Laboratory of Regional Climate–Environment for Temperate East Asia, Institute of Atmospheric Physics, Chinese Academy of Sciences, Beijing, China

**Correspondence:** Stéphane Vannitsem (stephane.vannitsem@meteo.be)

**Abstract.** We explore a methodology to statistically downscale snowfall – the primary driver of surface mass balance in Antarctica – from an ensemble of historical (1850-present day) simulations performed with an Earth System Model over the coastal region of Dronning Maud Land (East Antarctica). This approach consists in associating daily snowfall simulations from a polar-oriented Regional Atmospheric Climate Model at 5.5 km spatial resolution with specific weather patterns observed over 1979-2010 AD with the atmospheric reanalyses ERA-Interim and ERA5. This association is then used to generate the spatial distribution of snowfall for the period from 1850 to present day for an ensemble of ten members from the Earth System Model CESM2. The new dataset of daily and yearly snowfall accumulation based on this methodology is presented in this paper (MASS2ANT dataset, http://doi.org:10.5281/zenodo.4287517, Ghilain et al. (2021)). Based on a comparison with available ice cores and spatial reconstructions, our results show that the spatial-temporal distribution of snowfall is improved in the downscaled dataset compared with the CESM2 simulations. This dataset thus provides information that may be useful in identifying the large-scale patterns associated with the local precipitation conditions and their changes over the past century.

## 1 Introduction

In the context of the global climate change, the two polar ice sheets have increasingly gained attention, due to the threat of a massive sea level rise at the global scale (Garbe et al., 2020; IPCC AR6 , 2021). Over the last decades, the Greenland Ice Sheet has been melting at an increasing speed, both from the base and the surface (Mouginot et al. , 2019). In contrast, the Antarctic Ice Sheet is gaining mass at its surface through increased precipitations (Medley and Thomas , 2019) but the total contribution of the Antarctic Ice Sheet to the global sea level is also positive because of the melting at the base of the ice shelves due to the intrusions of warm water masses (Rignot et al. , 2019). It has been shown that the Antarctic snow accumulation increase is mainly explained by the increase in atmospheric humidity due to the rise of atmospheric temperature (Krinner et al., 2014;

Dalaiden et al., 2020). Therefore, the snow accumulation at its surface slightly reduces the positive contribution of the Antarctic Ice Sheet to the global sea level rise in the coming decades, thanks to the positive trend in precipitation.

The increase in snow accumulation has been documented regionally and evidenced to be the largest over the Antarctic Peninsula (Thomas et al. , 2008; Medley and Thomas , 2019). However, this increase is not observed everywhere on the Antarctic Ice Sheet (Medley and Thomas , 2019). Furthermore, studies in coastal Antarctic regions have revealed methodological limitations
in estimating the snow accumulation (defined as the difference between the total precipitation and sublimation, most generally called surface mass balance (SMB) (Lenaerts et al., 2019)). In particular, the tools and observations available have a too low spatial resolution in regions where snowfall is controlled by orography and spatial variations are high (Eisen et al., 2008). Dronning Maud Land (DML, 20°W – 45°E, map on Figure 2) is one of the sectors for which such uncertainties are large. As in the other Antarctic coastal regions, the gradient of snow accumulation follows the large scale topography, with maxima over
the coast and minima in the interior (Rotschky et al. , 2007). In addition to the large spatial scale, the snow accumulation is also highly variable at the scale of a few kilometers because of the ice rises and rumples that punctuate the coast (Lenaerts et al., 2014; Goel et al., 2020).

Over the last two centuries, the available reconstructions of Antarctic snow accumulation are deduced from the interpolation of the quality controlled ice snow accumulation records drilled at several locations (Favier et al., 2013). They assume an
averaged (Rotschky et al. , 2007) or a time-dependent spatial distribution (Medley and Thomas , 2019) or combine a climate model with ice core records using a data assimilation method (Dalaiden et al., 2021). However, those reconstructions also have large uncertainties too in regions with high spatial variability like DML. In addition to the ice core records, General Circulation Models (GCMs) provide another source of information for the snow accumulation over the past centuries. In contrast with ice core records, which provide estimations of snow accumulation for specific locations, climate models are
spatially complete and thus offer an Antarctic-wide estimation of snow accumulation. However, due to the low horizontal resolution of GCMs (typically 100km), the GCMs tend to underestimate the spatial variability of snow accumulation compared to observations (Palerme et al. , 2017). In order to better resolve the small-scale processes, regional climate models (RCMs) have been adapted over the polar regions (Mottram et al. , 2020) allowing a more detailed modelling of the processes that drive the snow accumuation variability at a scale of a few kilometers (Lenaerts et al., 2019). However, most simulations performed
with a RCM over the Antarctic Ice Sheet are only available from the satellite era (1979 afterwards). A 40-year period seems too short to estimate potential long-term trends or analyze the mechanism responsible for the the large interannual variability in snow accumulation (Medley and Thomas , 2019; Dalaiden et al., 2020; King and Watson, 2020).

Indications of a recent increase in snow accumulation have been found at some locations in Dronning Maud Land (Lenaerts et al., 2013; Schlosser et al. , 2016; Medley et al. , 2018; Philippe et al. , 2016). However, a stationary or decreasing trend has
been found elsewhere (Thomas et al. , 2017; Vega et al. , 2016; Altnau et al., 2015; Schlosser et al. , 2014), leading to large uncertainties on the net contribution of the region to the surface mass balance of the ice sheet. To address these uncertainties, we present a statistical method to downscale daily snowfall (the main contributor to snow accumulation (Lenaerts et al., 2019)) from an Earth System Model to 5.5 km over the 1850–2014 CE period, in order to better document and understand the long-term trend in the DML region along and the physical atmospheric mechanisms responsible for them.

Statistical methods can be set up to emulate the precipitation field, relating the spatial pattern of precipitation given by a RCM to large scale dynamics of the global climate model. We explored such a methodology to statistically downscale the dominant snow accumulation component over the area, the snowfall, from the climate model historical simulations (1850-present day). A method based on the search for analogs has been set up over a period of 32 years from the ERA-Interim reanalysis (1979-2010 AD; Dee et al. (2011)) in association with snowfall from the polar-oriented Regional Atmospheric Climate Model version 2.3 (RACMO2.3) at $5.5$ km spatial resolution over Dronning Maud in East Antarctica. The method is then applied to the period from 1850 to present day using an ensemble of ten historical simulations with an horizontal resolution of $1° \times 2°$ performed with the Community Earth System Model (CESM2, Danabasoglu et al. (2020)). This ESM has been widely used for studying the Antarctic precipitation, as CESM simulates relatively well the surface climate in Antarctica (Lenaerts et al., 2016, 2018; Fyke et al., 2017; Kittel et al., 2021; Dalaiden et al., 2020). The method enables to derive ten time-dependent spatial distribution of the snow accumulation over Dronning Maud Land at $5.5$ km resolution from the CESM2 ensemble members over the past 170 years. In addition, we provide the time series of the ten first principal components of each of the four atmospheric variables used and that can be exploited to characterize the regional synoptic situations. This dataset aims at providing an improved representation of detailed spatio-temporal patterns of snowfall over Dronning Maud Land. Firstly, this dataset allows for a more accurate comparison of long-term trends and temporal variability with local measurements (e.g., ice core snow accumulation records). Secondly, it enables a better understanding of the potential differences between estimates at small scale (such as the ones derived from ice cores) and the ones at the scale of the grid of a GCM. Finally, it makes it easier to associate synoptic meteorological conditions to snowfall events that determine the annual totals. In this paper, we present first the method and the data used, then the structure of the database, and finally the validation with ice core records and a discussion of the uncertainties.

## 2 Method and input data

Statistical downscaling follows a general scheme composed of three to four steps, involving several sources of information. In our case, at least three sources are required: 1) a GCM with coarse resolution but spanning a long time period, 2) a global reanalysis with coarse resolution spanning a shorter (recent) time frame, but of high quality, and finally 3) a regionally optimized RCM, with a finer resolution, used to dynamically downscale of the global reanalysis. The objective is to evaluate the relationship (called here Perfect Prog - PP -) between the reanalysis and the RCM fields and then to apply it to the GCM in order to downscale its results to the scale of the RCM. Applying the PP directly to the GCM turned out to be unsuccessful due to discrepancies between the GCM climatology and the reanalyses. Therefore, three essential steps (Figure 1) are envisaged: the setting of a PP between the reanalysis and the RCM, the definition of a correction scheme of the GCM with the reanalysis over a common time frame to produce an "emulated" GCM, and finally the use of the PP on the "emulated" GCM for the whole period of the GCM integration.

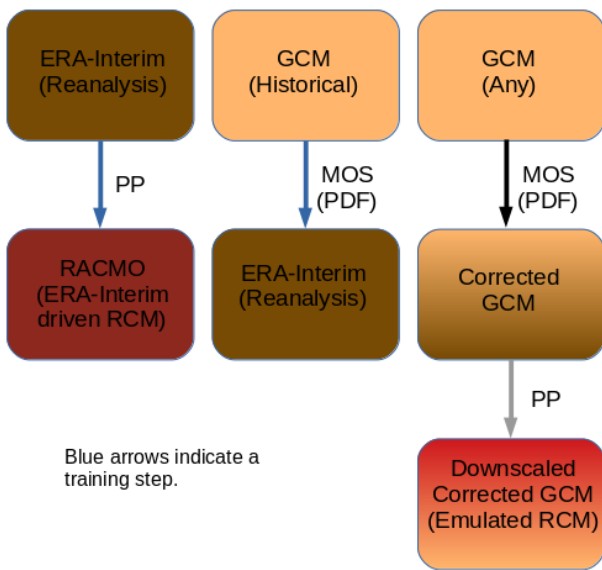

**Figure 1.** The statistical downscaling can be generally decomposed into 3 successive steps, including 1) the association of the reanalysis with the regional climate model by a Perfect Prog (PP, Maraun and Widmann (2018)), 2) a correction of the historical GCM to make it unbiased compared to the reanalysis using Model Output Statistics, and 3) the successive correction of the GCM followed by the application of the Perfect Prog, giving way to an emulated RCM estimation using the corrected GCM. Each uniform colour corresponds to a model (GCM, Reanalysis, RCM). The colours are mixed with a gradient for each operation including more than one source.

## 2.1 Input data

Four datasets are used in the development of the downscaled GCM: the reanalyses ERA-Interim and ERA5, the 10-member ensemble of CESM2 GCM simulations, and the optimized reanalysis-driven RCM integration RACMO2.3.

**The reanalyses.** ERA-Interim (Dee et al. , 2011) reanalysis has been largely validated all over the Earth surface, and also over Antarctica, where the precipitation was found to be consistent with satellites observations of CloudSat (Palerme et al. , 2017), and exhibits a significant correlation with the observed interannual variability (Wang et al. , 2016). Because of its relatively coarse spatial resolution (0.75°in latitude-longitude), it cannot resolve the atmospheric circulation over sharp local orographic changes. Therefore it cannot represent well the local scale precipitation over DML coast, which can be heavy due to orographic uplift (Palerme et al. , 2017). Heavy snowfall days ($> 10$ mm) from ERA-Interim correspond to less than 20% of the days identified by an automatic weather station near Kohnen (Welker et al. , 2014). By contrast, large-scale fields over the continent and Southern Ocean have been assumed to be more accurate. Therefore, ERA-Interim fields of relative humidity at 850, 700 and 500 hPa, geopotential height at 500 hPa, mean sea level pressure, sea ice extent, sea surface temperature and air temperature at 850, 700 and 500 hPa were extracted from ECMWF archives and eventually extrapolated. Snowfall was extracted the same way, for comparison. More recently, the ERA5 reanalysis (Hersbach et al., 2020) has been released with a

higher spatio-temporal resolution (0.25°and 1 hour). The same fields have been extracted from the Copernicus Climate Change Service Data Store (C3S CDS). The comparison of ERA5 snow accumulation with meteorological observations over the Antarctic Peninsula indicates an increased ability to identify strong precipitation events, which indicates a strong consistency between the synoptic weather patterns and the observed precipitation (Tetzner et al. , 2019).

**The Regional Climate Model.** The Regional Atmospheric Climate MOdel version 2.3 (RACMO2.3) was forced at its boundaries by the ERA Interim reanalysis, including an upper-air relaxation. This provides a simulation of the atmospheric variables and precipitations over the Antarctic Ice Sheet from 1979 to 2016 (Lenaerts et al., 2013; van den Berg and Medley , 2016; van Wessem et al. , 2016). An additional simulation at 5.5 km resolution (RACMO2.3p5.5) over specific regions of Antarctica, including Dronning Maud Land, allows to better represent the orographic effects of precipitation and the potential wind redistribution of snow. The latter configuration at higher resolution allows studying in more detail a region with complex surface topography and allows for a better comparison with SMB records (Lenaerts et al., 2018, 2014). RACMO2.3 has been recognized to have the best fit to recent AIS SMB observations compared to other atmospheric models and reanalyses (Wang et al. , 2016; Rignot et al. , 2019).

**The General Circulation Model.** The Community Earth System Model version 2 (Danabasoglu et al., 2020) provides a model framework that allows the reconstruction of the evolution of atmospheric variables at 1 to 2 degree resolution over the globe. A set of ten historical CESM2 runs using forcing from CMIP6 have been made available for the industrial period (1850 - 2014, Historical CESM2 CAM6). A different initialization has been generated for each member. The CESM2 CAM6 provides new historical runs issued from the latest development of the CESM model, and is shown to be an improvement over CESM1 (Danabasoglu et al., 2020).

## 2.2 The Method

The comparison of the meteorological and snowfall fields estimated from the high resolution (5.5 km) RCM RACMO2.3 and from the low resolution (regridded to 1°latitude-longitude) reanalysis ERA-Interim has revealed large local differences in the snowfall amounts and spatial distribution over the Antarctic coast in Dronning Maud Land. Two methods have first been unsuccessfully tried: First, we tried to find a relation between the origin of moist air masses and the intensity of associated high precipitation amounts. However, the analysis of back-trajectories (on-line HYSPLIT model applied on NCEP/NCAR reanalysis (Stein et al. , 2015)) from strong snowfall events on Dronning Maud Land has not identified a strong relation. This result contrasts with the success obtained with ERA-Interim and ERA5 in identifying the atmospheric rivers responsible of such events (Gorodetskaya et al., 2020). Secondly, another approach based on the Random Forest technique was tested, using as predictors several local atmospheric variables (air temperature, humidity, geopotential) or their large scale patterns. This method failed, probably due to the highly non-linear and non-systematic relations between large scale atmospheric patterns and snowfall intensity. In addition, since causal links can be model-dependent (Vannitsem et al. , 2019), a method that does not over-exploit those causal links is required to guarantee the transferability to other models in presence of large model uncertainties.

A downscaling method for snowfall based on the search for analogs has then been set up (Maraun and Widmann , 2018). The search for analogs bypasses the problem of non-systematic relations, as long as similar synoptic situations provide on average similar precipitation amounts. The choice of a daily time step is done in accordance with the duration of high precipitation events over Antarctica's coast, which account for about 60% of the annual snowfall (Schlosser et al. , 2010; Reijmer and van den Broeke , 2003; Turner et al. , 2019), last between 12 hours to 7 days long, and therefore can most of the time be associated to synoptic atmospheric situations (Reijmer and van den Broeke , 2003; Schlosser et al. , 2010; Welker et al. , 2014; Gorodetskaya et al., 2020). The meteorological fields identified to be the most explanatory (and possibly replicable by climate models) for precipitation rates over the Antarctic continent have been decomposed in Empirical Orthogonal Functions (EOFs). The principal components weights (PCs) of the dominant EOFs (the first ten) are used for the selection of the analogs (Sneyers and Goossens , 1988; Hannachi , 2004). To select the large-scale meteorological fields of interest for the downscaling method, we have built for a set of 50 points from the RACMO2.3 domain an analog database from the association between the principal components weights (PCs) of different fields from ERA-Interim and the RACMO2.3 precipitation over a 11 years period (1979-1990). These points correspond to the highest annual snowfall accumulation over the area. The fields tested were: geopotential height at 700 and 500 hPa, air temperature at 500, 700 and 850 hPa, relative humidity at 850, 700 and 500 hPa, surface atmospheric pressure, sea-ice cover, and total precipitation (liquid and solid). One to four fields were used in the different tests. The statistical evaluation of the performance was based on correlation and root-mean square difference at both daily and annual scales after a bias correction (quantile mapping). The analysis over the validation period (1991-2000) indicates that the optimal choice is reached with four fields: 1) geopotential height at 500 hPa, 2) surface pressure, 3) relative humidity at 700 hPa and 4) total precipitation. The same evaluation methodology was used to select 1) the optimum number of closest analogs used for the estimation, namely 20 analogs, 2) the way to select the best estimation from the distribution of analogs: the mean of the ensemble seems the most appropriate, 3) the optimum minimum geographical spanning of the reanalysis fields: latitude 40°S-90°S, longitude 10°W-60°E, and 4) the length of the training period: 3 is a minimum, but we use 11 years without a large statistical gain. As expected, the spatial and temporal variations of RACMO2.3 are preserved (Figure 2 shows an example of time series comparison and a comparison over Princess Ragnhild Coast characterized by the presence of ice rises near the coast). The root mean square difference on a daily basis over the validation period ranges from 10% in large accumulation areas near the coast to 15% in the inner regions. The advantage of the analog method is that it allows one to identify the major types of weathers delivering precipitation and their occurrence over time in the form of PCs. This could be of interest for the analysis of a frequency change in the precipitation over long time periods.

The downscaling method was then repeated over the ensemble of the ten climate runs from the CESM2 model over the period 1850-2014. Before the downscaling could be applied, we first needed to make the PCs compatible with the reanalyses. A simple bias correction based on the linear regression of EOFs (Feudale and Tompkins , 2011; Yu et al, 2018), followed by a quantile mapping, transforms the CESM2 original PCs into "Reanalysis-like" PCs. The operation is done after the verification of the similarities of the EOFs among the members. The principal components can then be compared to the analog database in search for the closest events. As during the training process, quantile mapping defined over a 10-year period with the reanalysis is applied to the complete time series to obtain the final downscaled estimations (Figure 3). No significant high differences are

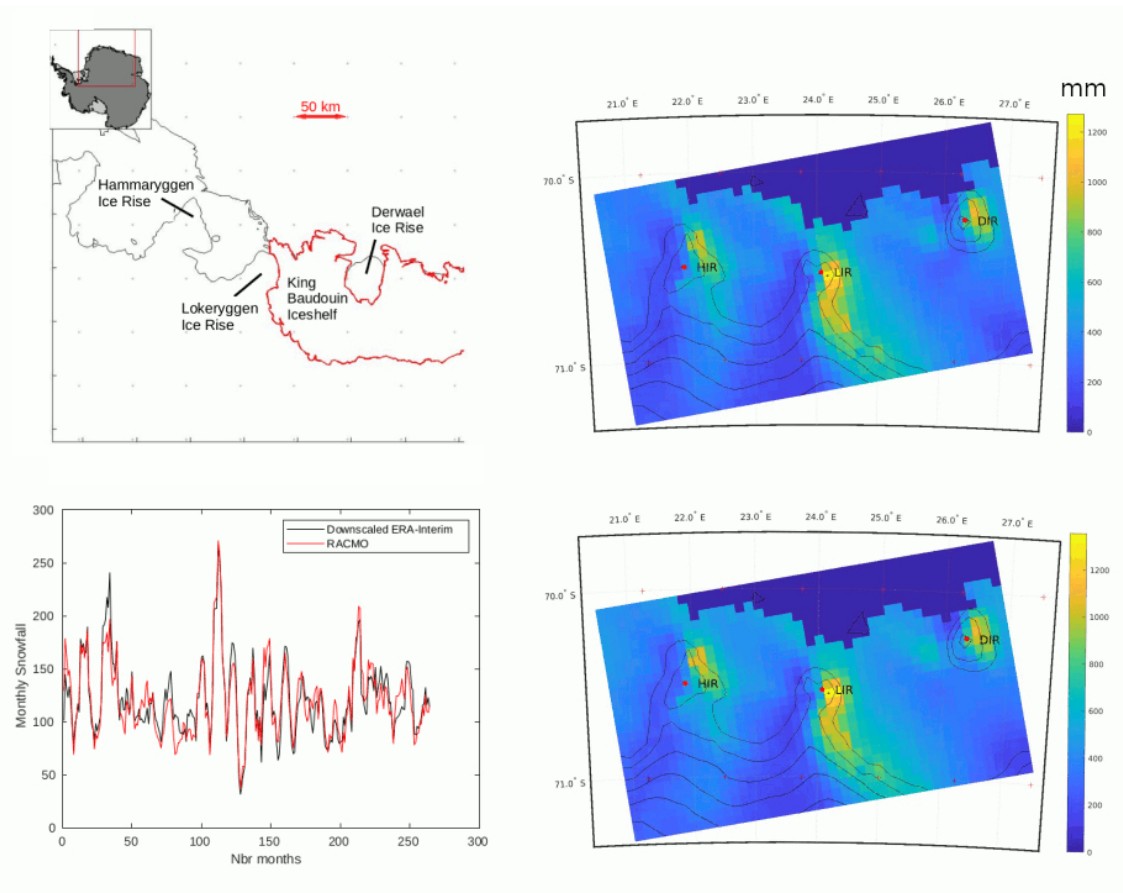

**Figure 2.** The map shows the Princess Ragnhild Coast, which is part of Dronning Maud Land (red box), and the whole Antarctic continent. Verification of the performance of the downscaling scheme trained on 11 years (1979-1990) of ERA-Interim data and RACMO2.3 daily snowfall and applied on ERA-Interim for a 10 years period (1991-2000). The monthly time series comparison over a location of maximum of annual snowfall accumulation shows a high degree of accuracy, while the spatial comparison of the accumulated snowfall (in mm) over 1996 on Princess Ragnhild Coast in presence of ice rises (eg Derwael Ice Rise – DIR -, Lokeryggen Ice Rise – LIR and Hammaryggen Ice Rise - HIR) illustrates the high degree of fidelity of the analog method (top right) in reproducing the RCM (bottom right) accumulation patterns, especially in the West-East difference of accumulation around ice rises (Kausch et al., 2020).

expected from one member to another, as the initializations have been carefully controlled and the spectral analysis of the time series of PCs give very similar results (Danabasoglu et al., 2020).

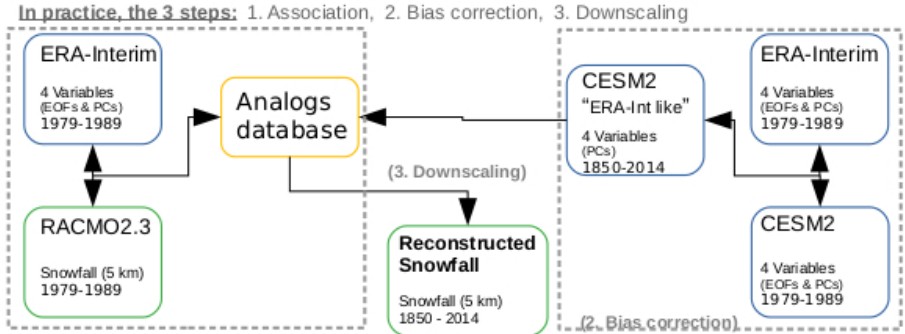

**Figure 3.** The analogs method follows 3 steps: 1) the building of the analogs database using the association of the reanalyses PCs to the snowfall accumulation from RACMO2.3, 2) the transformation of CESM2 into "reanalysis like" fields, 3) the search in the analogs database. The second step ensures the compatibility of the principal components between the database and the GCM.

## 3 The dataset

### 3.1 Structure of the dataset

The dataset is composed of 46 files (netcdf4 format) in total, which accounts for about 500GB, when uncompressed. The yearly accumulated snowfall maps are provided in polar stereographic projection at 5.5 km resolution, conforming to the RACMO2.3p5.5 run set-up: 1 file per downscaled CESM2 run, for each re-analysis used for training (20 files). The daily dataset is provided in the form of time series (20 files): 10 files (netcdf4 format), each containing the time series of daily snowfall accumulation over each grid point of the RACMO2.3 domain on the ice sheet (DML coastal region) for the period 1850-2014, 365 days per year. Each file corresponds to a downscaling of one of the 10 runs from CESM2, based on the training on ERA-Interim and RACMO2.3. Latitude, longitude, corresponding line and column in the RACMO2.3 domain are stored for each grid point. Extraction of time series for a specific location is therefore straightforward, and, thanks to the column and line, it is possible to recompose daily maps. Due to the size of the daily files, only a set of 2 files are stored along with the annual data files on Zenodo, an open-access repository operated by CERN, the European Organization for Nuclear Research, and the whole set is available on request. Another set of 10 files with the same structure is stored with the results from the application of the same method trained on ERA5. In addition the time series of the 40 principal components used are stored in separate files, one for ERA-Interim, and one for ERA5 trained simulations (Figure 4), all provided with embedded metadata. PCs time series for the 40 EOFs for all the CESM2 members are stored in 2 additional files. The EOFs maps from ERA-Interim and ERA5 are stored in 2 separate files. At last, PCs time series from ERA-Interim and ERA5 fields are stored in 2 files.

### 3.2 Examples of maps (annual, daily)

As expected, daily fields differ among the different members of the downscaled CESM2 results. When a high precipitation event over DML can be present on a specific day in member 7, it may not be the case for all the other members (Figure 5).

# MASS2ANT Database
Snowfall 5 km over DML, Antarctica

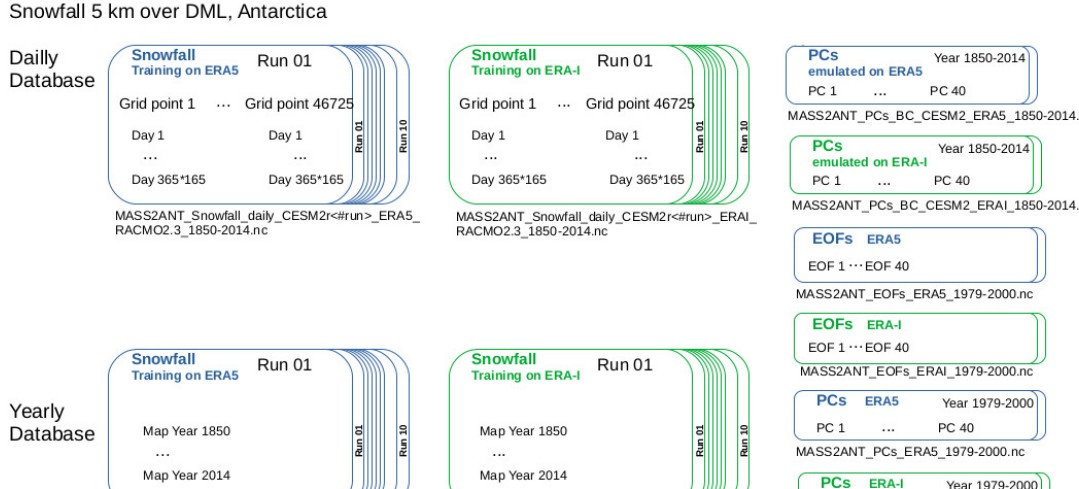

# MASS2ANT Database
Snowfall 5 km over DML, Antarctica

| Metadata | | | | | | |
|---|---|---|---|---|---|---|
| Global Attributes | | | Dataset | Dataset 1 | Name | Annual snowfall accumulation |
| | Authors | Ghilain N., Vannitsem S., Dalaiden Q., Goosse H., De Cruz, L. | | | Unit | mm.w.e. per year |
| | Institution 1 | Royal Meteorological Institute of Belgium | | | Date start | 01-Jan-1850 |
| | Institution 2 | UCLouvain, Eath&Life Institute | | | Date end | 31-Dec-2014 |
| | Projection | Lambert equidist pole rotated | | | Time step | 1 day / 365 days |
| | Method | Analog based on EOFs (Z500hPa, SP, Rh700hPa, Total Precip) | | | Source RCM training | RACMO2.3 |
| | Domain | Dronning Maud Land, Antarctica | | | Source GCM training | ERA5 / ERAI |
| | Spatial resolution | 5 km | | | Source GCM evaluation | CESM2 |
| | Version netcdf | | | Dataset 2 | Name | Latitude |
| | Date production | 15-Nov-2020 | | | Unit | Degree |
| | Release reference | 1.0 | | | Source | RACMO2.3 |
| | Code Source | Matlab/Fortran90 | | Dataset 3 | Name | Longitude |
| | | | | | Unit | Degree |
| | | | | | Source | RACMO2.3 |

**Figure 4.** The MASS2ANT database (NetCDF files) includes 2 files for yearly accumulated snowfall (10 members each), and one per reanalysis used for training. The daily estimations are stored per year and per reanalysis used for training. In addition, PCs time series for the 40 EOFs for all the CESM2 members are stored in 2 separate files, as well as the 40 EOFs of the emulated CESM2. Embedded metadata include units, version geographical coordinates, and temporal stamps.

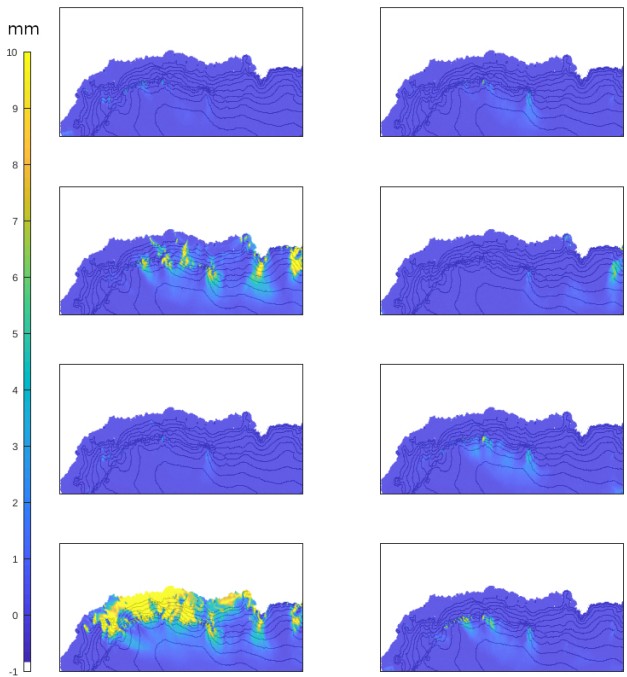

**Figure 5.** Recomposed maps of downscaled daily snowfall (in mm) over DML on the 1st day of 1850 for members 1 to 8. A saturation at 10 mm per day has been defined for visualization.

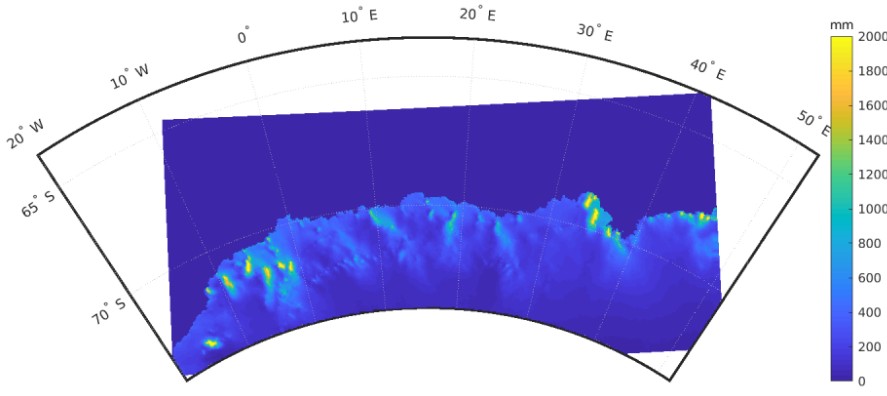

**Figure 6.** Map of downscaled snowfall accumulated over a year (CESM2 member 1). A saturation at 2000 mm has been imposed for a better visualisation.

While a large disparity is observed on daily maps, the effect is much smaller on the annual accumulation (Figure 6). We illustrate this by a daily map (members 1 to 8) and an annual accumulation over the entire domain of the downscaled CESM2 member 1, trained with ERA5 (Figure 5).

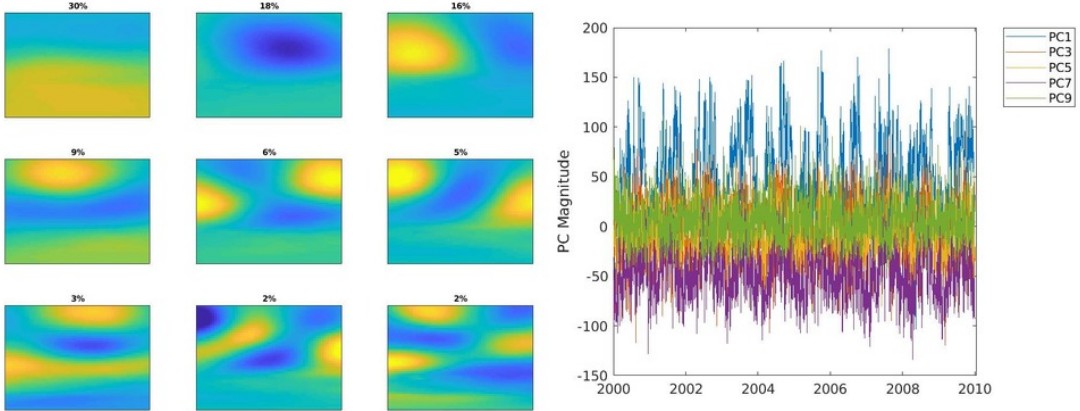

**Figure 7.** The configurations of the 9 first EOFs for surface atmospheric pressure with their relative occurrence are ordered by importance in the decomposition of the variability of the field (represented here unitless, covering the total area 40°S-90°S, 10°W-60°E). The time series of the associated PCs for one year show the different amplitudes.

For each date, the PCs are provided, and associated to the EOFs for each meteorological field. An analysis of the association between snowfall regimes with the synoptic patterns is then possible. In Figure 7, we show an illustration of the first 9 EOFs for Surface atmospheric pressure over the GCM domain considered (40°S-90°S, 10°W-60°E), as well as the time series of the associated PCs, showing their various amplitudes.

### 3.3 Comparison with ice core measurements

The annual variation of the surface mass balance is estimated from ice cores retrieved from a set of locations over the coastal DML. The length of the records differs between the different sites and can cover more than 150 years. Specifically, the annual accumulation computed from the downscaled daily snowfall is compared to SMB from the different ice core records from Thomas et al. (2017), after a conversion of ice core estimates to water equivalent height (Figure 1, Figure 8). A good overall match between the downscaled snowfall and the ice cores is found, as well as a large bias reduction compared to the use of CESM2 without downscaling.

We also briefly investigate the multidecadal trends and their uncertainties. Specifically, we have computed the trend in yearly snowfall accumulation for the period 1988 to 2014, relative to the mean accumulation around 1988 for each ice core site, and for each downscaled member. The result is an ensemble of 10 trends per site. For the 6 first sites, the ensembles show in average no trend, with individual estimate ranging from -0.7% to 0.7% (except for one member which exceed 1%for S100, S20 and DML16). For the 2 last sites (H72 and DIR), the range is larger: from -0.6% to 1% (except for one member reaching 1.5%). For the trends over 1850 to 2014, the estimates are ten times smaller for the 5 first sites. At DML16, H72 and DIR, 9 out of 10 downscaled estimates give a positive trend ranging from 0 to 0.2% (DML16 and H72) and from 0 to 0.3% (DIR).

**Table 1.** Ice cores drilled over Dronning Maud Land used for comparison, extracted from Thomas et al. (2017)

| Ice Core | Latitude | Longitude | Start date | End date | Reference |
|---|---|---|---|---|---|
| S100 | -70.24 | 4.80 | 1850 | 1999 | Karczmarska et al. (2004) |
| S20 | -70.25 | 4.82 | 1956 | 1996 | Isaksson et al. (1999) |
| H72 | -69.20 | 41.08 | 1850 | 1999 | Nishio et al. (2002) |
| B04 | -70.62 | -8.37 | 1892 | 1981 | Schlosser (1999) |
| DIR | -70.25 | 26.34 | 1850 | 2011 | Philippe et al. (2016) |
| E91 | -73.60 | -12.43 | 1932 | 1991 | Isaksson et al. (1996) |
| DML16 FB9813 | -75.17 | 5.00 | 1850 | 1997 | Oerter et al. (2000) |

## 3.4 Comparison with other sources and other studies

Almost all the available datasets of spatial estimates of snowfall accumulation or SMB over Dronning Maud Land are 1) at high resolution (20-30 km (Lenaerts et al., 2018; Agosta et al., 2019)) but spanning the last 30 years, 2) spanning a longer period but at low resolution (0.75°, Medley and Thomas (2019)), or 3) at high resolution but averaged in time (Rotschky et al. , 2007).

Over other regions, downscaling using a physical orographic model has been used to obtain maps at high resolution from a RCM over a longer period of time (Agosta et al., 2013). In this study, we compare our dataset to two other sources. The first reconstruction covers Western Dronning Maud Land (Rotschky et al. , 2007), which offers a static view of the accumulation patterns based on the spatial interpolation of ice core data (data available from the PANGAEA website http://doi.pangaea.de/10.1594/PANGAEA.472297). A visual comparison with the CESM2 member 1 downscaled (trained with ERA5) shows an

enhanced variability along sharp elevation changes (Figure 9). The second reconstruction over the entire domain is based on ice core datasets and climate patterns from reanalyses over the last 150 years on a yearly basis (Medley and Thomas , 2019), of which we only consider the reconstruction using ERA-Interim (data available at https://earth.gsfc.nasa.gov/cryo/data/antarctic-accumulation-reconstructions). The downscaled CESM2 member 1 averaged over the total period seems in agreement with the averaged reconstruction based on the ice core records, especially to reconstruct the gradient from the coast to inner land, but

depicts much more details related to topography (Figure 10).

## 4 Uncertainties

The uncertainties in the daily snowfall estimates compiled into this dataset can arise from 1) the choice and accuracy of the model sources (regional climate model, re-analysis, "historical" climate model), and 2) the accuracy of the downscaling method (choice of parameters, choice of predictive variables). In this section, we report a (non-exhaustive) quantification of

the uncertainty level related to both the choice of model sources and of the parameters of the method.

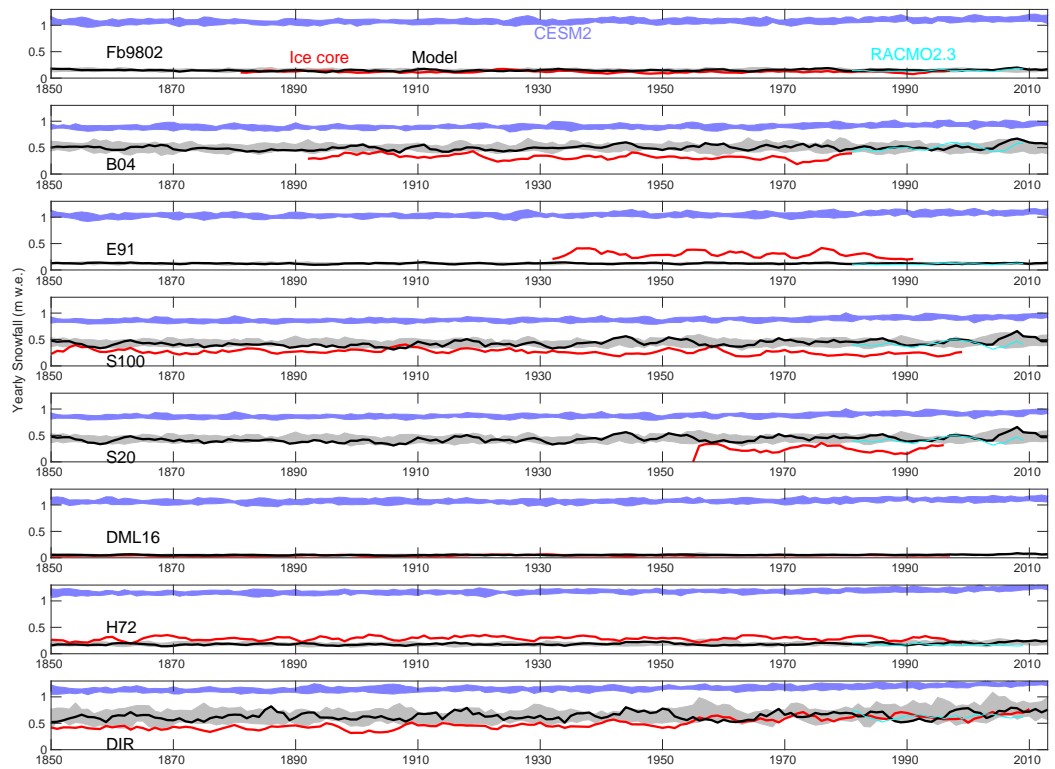

**Figure 8.** The time series of yearly accumulated snowfall extracted from the database (CESM2 member 1, training with ERA-Interim, and all members in shaded gray) is compared to the SMB estimated from ice cores available from (Thomas et al. , 2017) (Ice core data available on the webpage of PAGES2k/Antarctica2k and in https://data.bas.ac.uk/full-record.php?id=GB/NERC/BAS/PDC/00940). RACMO2.3 time series has been superimposed (cyan) and the 10 CESM2 simulations (shaded blue), showing the large bias reduction thanks to the use of RACMO2.3 for the training. A 4-year moving average has been applied on the time series for the visualization.

## 4.1 Sampling in choice of analogs

A simple way to assess the effect of the sampling of the analogs on the estimation uncertainty is to use the bootstrap techniques. We have created a distribution of 100 ensembles of analogs at the ice core sites, with the 20 analogs randomly drawn with repetition (Figure 11). The downscaled CESM2 member 1 with 100 different analogs draws represented at the ice core sites. 235  The uncertainty is between 5 and 10 % of the mean value.

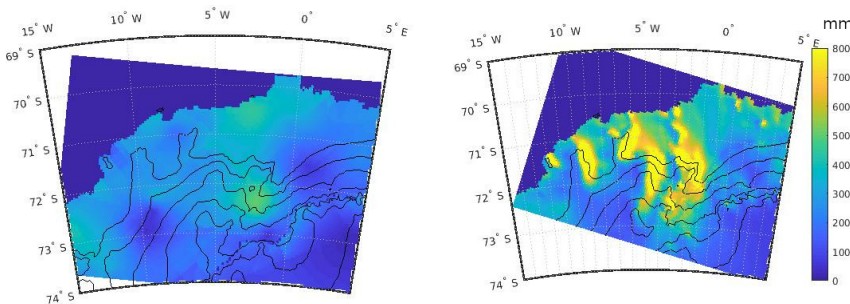

**Figure 9.** The visual comparison of the average annual snowfall accumulation over Western Dronning Maud Land between the database from (Rotschky et al. , 2007) and the downscaled CESM2 member 1 (trained with ERA5) shows more pronounced asymmetrical patterns near sharp surface elevation changes. The difference in magnitude should be taken with caution, as only the snowfall component is provided by the MASS2ANT database, while (Rotschky et al. , 2007) is the integration of the total surface mass balance.

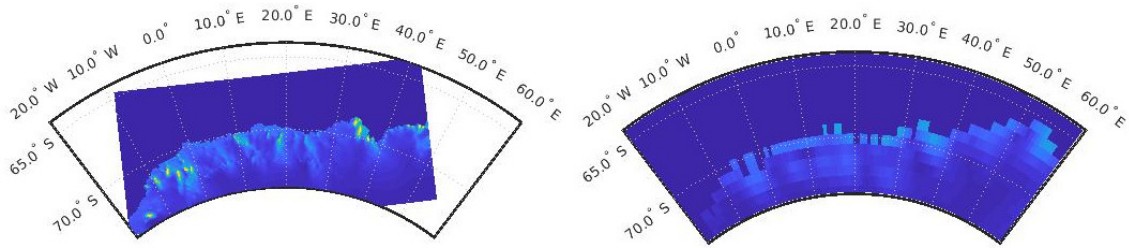

**Figure 10.** The coast-to-inland gradient of the downscaled CESM2 member 1 averaged over the total period seems in agreement with the averaged reconstruction based on the ice core records of (Medley and Thomas , 2019), but depicts much more details that could be useful in detailed analysis of ice core records representativity (same color scale).

## 4.2 Choice of reanalysis used for training

The choice of reanalysis used for training the analogs database has an effect on the downscaling obtained from CESM2, as there still remain biases in reanalyses, eg a dry bias in the interior of the ice sheet (Bromwich et al., 2011). The choice of reanalysis influences the result, at equivalent training scores (Figure 12). This is one of the reasons for extending the database to ERA5 as training reanalysis instead of ERA-Interim. More independent reanalyses could be used to further assess the uncertainty. The difference CESM2 downscaling using ERA-Interim or ERA5 in the training reveals an uncertainty of 25% in average at the ice core sites.

## 4.3 Choice of "historical" GCM runs

CESM2 simulations consist of 10 members, corresponding to different initializations (Danabasoglu et al., 2020). The inherent model uncertainty is therefore impacting the results and is quantified here (Figure 13). The scatter of the 10 members of

**Table 2.** Summary of the uncertainty levels associated to the choices of input model or sampling method. The uncertainties (mean uncertainties) are estimated on yearly accumulation of snowfall, relative to the basic methodology followed to derive the database, for at least one time series over the domain.

| | Snowfall Uncertainty (% mean yearly accumulation) |
|---|---|
| Sampling in choice of analogs [1] | 5 to 10 % |
| Choice of reanalysis for training [2] | 25% |
| Choice of "historical" GCM runs [3] | 17-22 % (std) to max 30-38% |

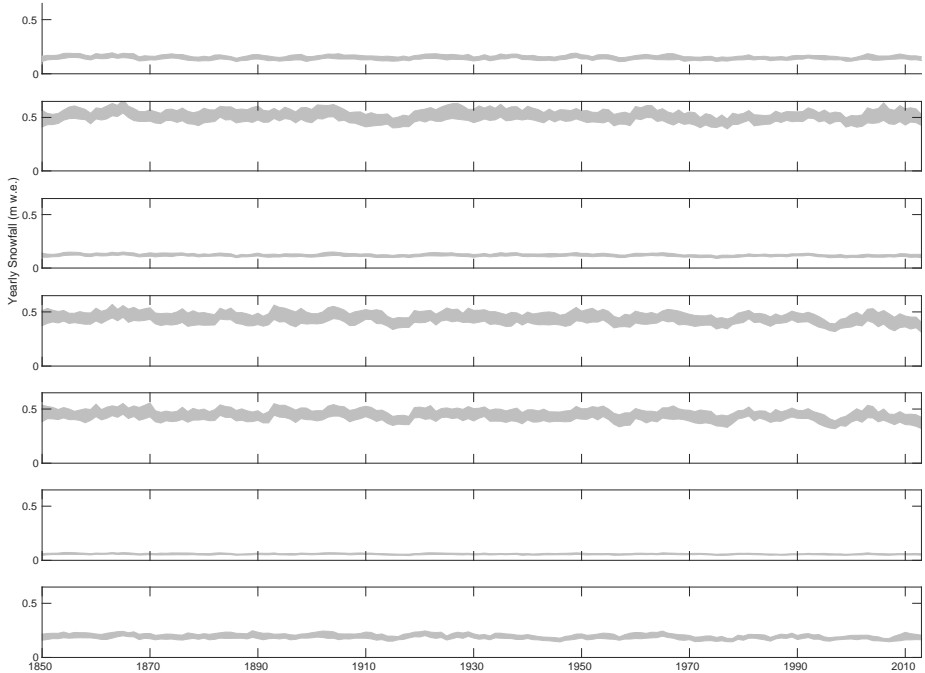

**Figure 11.** The downscaled CESM2 member 1 with 100 different analogs draws represented at the ice core sites. The uncertainty is between 5 and 10 % of the mean value.

CESM2 downscaling using ERA-Interim in the training reveals an uncertainty of 20% (standard deviation) and up to 40% at some ice core sites, maybe a direct consequence of the CESM2 members not being temporally correlated.

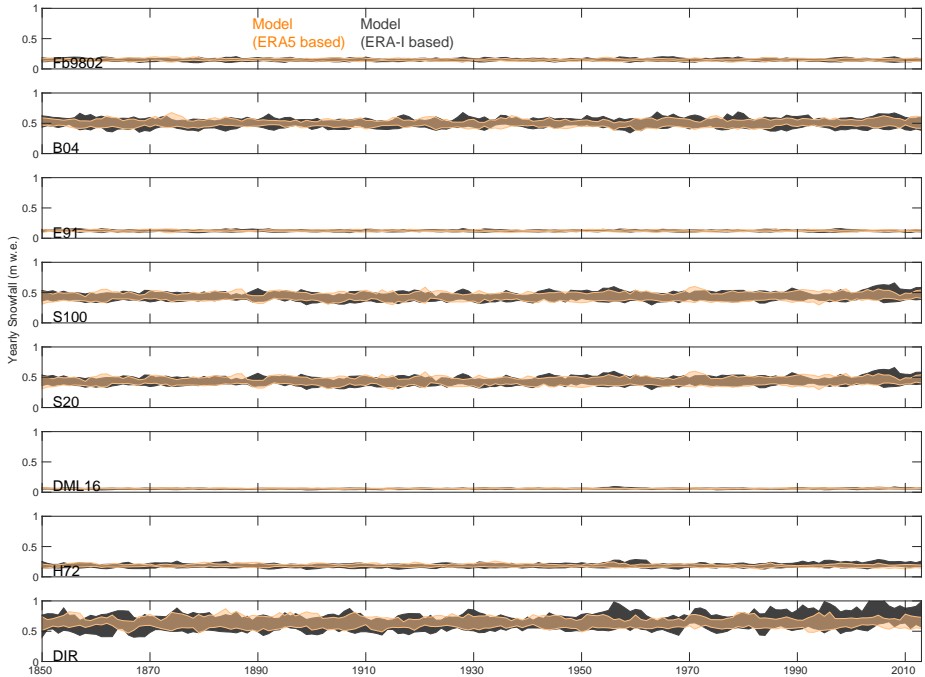

**Figure 12.** The difference CESM2 downscaling using ERA-Interim or ERA5 in the training reveals an uncertainty of 25% on average at the ice core sites.

## 4.4 Choice of RCM model

At last, the choice of the RCM critically drives the dynamical link between large-scale patterns and local snowfall, and the average and maximum amplitude of snowfall accumulation. An assessment of the uncertainty caused by the choice of RCM could be done using another RCM, with equally proven quality. For example, the Modele Atmospherique Regional (MAR) is another recognized RCM used for polar climates at high resolution. The newest simulations over Antarctica at 30 km resolution (Agosta et al., 2019) have shown similarities with RACMO2 at 27 km resolution, but revealed localized differences over the ice sheet related to a more realistic sublimation of falling snowfall in comparison with RACMO2 (Gallée et al., 2013). A simulation at the same resolution with another RCM optimized for polar regions, like MAR, could be of great interest to better frame the uncertainty linked to RCM physics. However, the uncertainty linked to the choice of RCM has not been evaluated in this study and could be envisaged to extend the database if new RCM simulations at such resolution are available in the future.

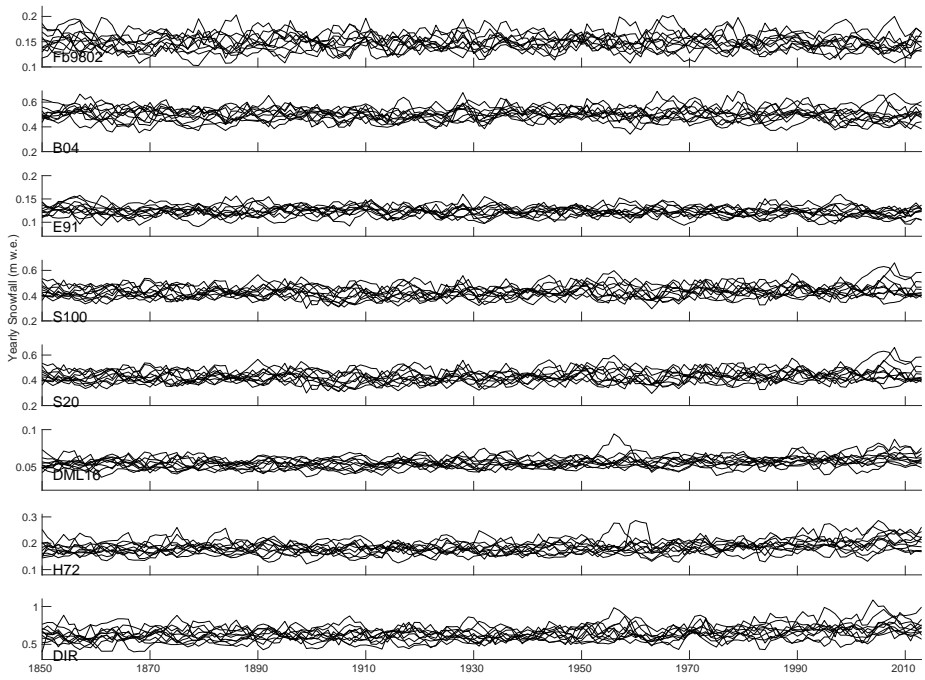

**Figure 13.** The scatter of the 10 members of CESM2 downscaling using ERA-Interim in the training reveals an uncertainty of 20% (standard deviation) and up to 40% at some ice core sites.

## 5  Conclusions

We propose here a reconstruction of snowfall evolution over Dronning Maud Land, Antarctica, at $5.5$ km resolution using
an analog-based downscaling technique. This technique has allowed us to exploit the detailed spatio-temporal estimation of snowfall from 30-year RACMO2.3 simulations in combination with synoptic patterns from recent reanalyses to statistically downscale the historical runs from CMIP6 (CESM2 model). The resulting database stores the ensembles of daily accumulated snowfall from 1850 to 2014, the pertinent information for synoptic patterns analysis (the principal components weights and empirical orthogonal functions from four large-scale meteorological fields), and the annual evolution of accumulated snowfall
over Dronning Maud Land. The database can be used to analyze the detailed contribution of snowfall to the surface mass balance over the region, its evolution and its association to synoptic weather conditions. The method can be easily replicated with new RCM and GCM simulations.

# 6 Data availability

The files of the dataset (the annual snowfall, the PCs and the EOFs) are available on Zenodo platform (http://doi.org:10.5281/zenodo.4287517). However, due to size limitations, only 2 daily snowfall files out of 20 have been stored there, the whole set is available on request to the contact author.

*Author contributions.* Project proposal and funding: SV & HG; Scientific supervision: SV; Design of the method: LDC, HG, SV & NG; Climatological Analysis NG, WW & SV; Implementation, tests, method tuning, verifications, database creation, validation: NG & SV; Analysis of the results: all authors; Preparation data material: QD & NG. Manuscript drafting: NG; Manuscript revisions: all authors.

*Competing interests.* The auhors declare that they have no conflict of interest.

*Acknowledgements.* The authors thank the Santander Meteorology group for making available the MeteoLab Toolbox Matlab software (Cofino et al., 2013), which was re-used and partly re-coded in fortran90 using multi-core parallel computing for the processing of the database. We thank Chad Greene for providing the Climate Toolbox for Matlab, which was used to verify intermediate steps. We thank Dr Jan Lenaerts for providing the RACMO2.3 fields over DML Dr Marie Cavitte for suggestions on the use of ice core datasets, and Dr Jean-Louis Tison, Dr Stef Lhermite and Sarah Wauthy for fruitful discussions. The study was funded in the framework of the MASS2ANT project (https://www.elic.ucl.ac.be/users/klein/Mass2Ant/index.html, contract No BR/165/A2) by the Belgian Science Policy.

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
