# Peer review of "Large ensemble of downscaled historical daily snowfall from an Earth System Model to 5.5km resolution over Dronning Maud Land, Antarctica"

_Earth System Science Data, 2021_

## Referee Comment (RC2)

Earth System
Science
Data

[referee-annotated manuscript omitted]

---

## Author Comment (AC2)

**Reply to Reviewer 2's comments**

**Reviewer 2: The paper estimates annual precipitation rates over the coastal area of Dronning Maud Land. Using a downscaling from general circulation models of the atmosphere, a regional model and reanalysis data, the paper attempts to identify key atmospheric processes predicted back in time and which can be used to infer surface precipitation rates. The idea is intriguing and worthy of study. It is consistent with the theme of this journal.**

*Authors:*

*The authors thank the reviewer for the careful reading of the manuscript and the comments.*

**Reviewer 2: The paper requires editorial revision and further analysis. Specifically, the paper repeatedly states that the results are an estimate of surface mass balance. In fact and as is also stated, the estimate is surface precipitation rates. Further reference to Lenaeerts et all 2019 and the magnitude of other contributions to SMB would be useful.**

*Authors:*

*Over Antarctica, SMB variability has been shown to be mostly driven by snowfall, with a contribution often one to two orders of magnitude larger than the other components (sublimation, melting, runoff, snow ablation by wind). Lenaerts et al (2019) will be cited in the introduction, with estimates of the order of magnitude of the different components. This very strong link between precipitation and SMB has to be noted, because ice cores provide measurements of the SMB, and precipitation is a good proxy for the comparison with those measurements. However, we will carefully revise the manuscript to make a clear distinction when SMB can be mentioned, and when precipitation should be used instead.*

**Reviewer 2: Section 3.1 on the dataset should go at the end of the paper once a convincing argument about the utility of the data set has been made. To that point, the variation between model and ice core data seems to be about 25 % (Figure 8) and it needs to be made clear this is a meaningful result. Moreover and perhaps more importantly, there does not seem to be a strong correlation between the variations in the model data and the ice core data. If there is a correlation it should be quantified. I expected to read a conclusion about long term trends in precip rates as hinted at in the introduction. I did not find that nor do I think such a conclusion is possible based on inspection of Figure 13. More discussion on that point would be useful and would strengthen the paper whether a trend is discernible or not.**

*Authors:*

*We propose to maintain the information provided in Section 3.1 at the beginning of section 3, as in the submitted manuscript. It describes what information is accessible. We believe that this journal is precisely oriented toward the description of such a dataset and is thus at the core of the paper. The rest of the section illustrates the different parts of the accessible dataset that can be of interest for further analyses. The motivations (and utility) of the dataset will be clearly re-stated at the end of the introduction. Variations and correlations compared to ice core data have been computed for each member of the downscaling, and will be summarized in the text. However, the comparison should be taken cautiously, as there is still a mismatch between the spatial resolution and representativity of the dataset and the ice cores. Multi-decadal trends have also been computed for each time series, and will be discussed in the revision.*

**Reviewer 2: I made editorial comments in the text. My recommendation is to return the paper to the authors and to strongly encourage them to make major revisions before resubmitting. I believe they are on the right track to an interesting result but it needs a bit more work**

*Authors:*

*Thank you for the comments on the manuscript.*

- *As said, reference to SMB and precipitation will be carefully checked in the text.*

- *Inconsistencies in the introduction will be checked, eg ice shelf/ice sheet.*

- *Snowfall is derived from CloudSat: SMB will be replaced by snowfall*

- *We will rephrase l.108-115, so as to better express the first trials and why they failed, to justify the method used.*

- *Zenodo will be explained: "Zenodo, an open-access repository operated by CERN".*

- *We will ensure that there is no confusion in the text between CESM member and downscaled member.*

---

## Author Response (AR1)

**Reply to Reviewer 1's comments**

**General**
**This is an interesting application that remaps coarse-resolution Antarctic accumulation fields from a GCM to high (5.5 km) resolution using two intermediate products, a coarse resolution re-analysis product (ERA-Interim/ERA5) and a high-resolution regional climate model (RACMO). There are three problems that I identified while reading the paper: the introductory sections are poorly written, the goal is not well-motivated, and most of the figures are of mediocre quality.**

*Authors:*

*The authors thank reviewer 1 for careful reading of the manuscript and the comments.*

**Major comments**
**Unfortunately, the introductory parts of the paper are not well written. Many formulations could be clearer and more concise. Sometimes not even the meaning of a sentence is clear, for example, the sentence starting at l. 19: "While the Greenland Ice sheet is eroding at an increasing speed both from the base and the surface, the Antarctic Ice Sheet is sometimes viewed as subject to a mitigation mechanism to the observed melting of the ice shelf through an increased coastal precipitation due to a higher atmospheric humidity." is really unclear. It is true that in a warmer atmosphere, snowfall over ice sheets is expected to increase, mitigating future mass loss. This is true for both Greenland and Antarctica, so why make the distinction? Solving this requires a thorough and critical re-reading of the manuscript, maybe the co-authors can be of assistance here?**

*Authors:*

*The introduction has been rewritten and carefully reviewed by all the co-authors to ensure that all the text is clear and concise. Recent references have been added and the objective of the paper as well as the added value of the database with respect to existing datasets have been clarified.*

**Please better motivate the goal of the study. The title suggests that 'real' snowfall amounts are reproduced, but these are merely time series with improved statistics that do not represent 'real' events, rather 'real' variability. So please clarify: what are these data derived for?**

*Authors:*

*The authors thank reviewer 1 for bringing to our attention the ambiguous meaning of the title. As said by reviewer 1, this reconstruction is not a "reanalysis" of daily snowfall ('real daily snowfall'), but a reconstruction of daily rates of snowfall over a long period with spatio-temporal characteristics similar to the ones of the highest resolution products currently available. As such, it presents an improvement with respect to available information given by reanalyses & RCMs (short time period, max 40 yrs) and climate GCMs (coarse resolution, 1 degree). The reason for producing this dataset is to provide an improved representation of detailed spatial patterns of snowfall over Dronning Maud Land. Firstly, this dataset allows for a more accurate comparison of trends and long-term temporal variability with local measurements (here, SMB deduced from ice cores). Secondly, it enables a better understanding of the potential differences between estimates at small scale (such as the ones derived from ice cores) and the ones at the scale of the grid of a GCM. Finally, it makes it easier to associate synoptic meteorological conditions to snowfall events that determine the annual totals.*

*We therefore suggest to modify the title as follows, to emphasize that the primary source of information is a climate model:* Large ensemble of downscaled historical daily snowfall from an Earth System Model to 5.5km resolution over Dronning Maud Land, Antarctica.

*The motivations have been made clearer in the different parts (abstract, introduction and conclusion), especially at the end of the introduction by re-stating explicitly the purpose and assets of the dataset as detailed in the response above.*

*l.67-72: "This dataset aims at providing an improved representation of detailed spatio-temporal patterns of snowfall over Dronning Maud Land. Firstly, this dataset allows for a more accurate comparison of long-term trends and temporal variability with local measurements (e.g., ice core snow accumulation records). Secondly, it enables a better understanding of the potential differences between estimates at small scale (such as the ones derived from ice cores) and the ones at the scale of the grid of a GCM. Finally, it makes it easier to associate synoptic meteorological conditions to snowfall events that determine the annual totals."*

**Reviewer 1: Please improve figure quality, below are some suggestions.**

**Fig. 1: what do the colours of the text boxes represent?**

**Fig. 2: add elevation contours to map, what are axis units? To see the improvement, time series should also show the non-downscaled ERA-Interim time series. Increase font size, add units to colour bar, 'nbr months' is not really a clear axis label etc.**

**Fig. 5: increase the font size, add a unit to the colour scale.**

**Fig. 7b: unclear which colour represents which principal component**

**Fig. 9: increase the font size, add a unit to the colour scale.**

**Fig. 10: add colour scale.**

**Fig. 11, 12, 13: consider removing.**

*Authors:*

*Thank you for the suggestions to improve the readability of the figures.*

*Fig 1: Each uniform colour corresponds to a model (GCM, reanalysis, RCM), we mix colours with a gradient when an operation including more than one source has been done. This is made explicit in the caption. (p.4: "Each uniform colour corresponds to a model (GCM, Reanalysis, RCM). The colours are mixed with a gradient for each operation including more than one source.")*

*Fig 2: We have increased the font size and the units have been added to the colour bar. The non-downscaled ERA-Interim data has not been added, as the purpose of the figure was to show the verification of correspondence between the RCM and the trained method..*

*Fig 5 & 9: the font size has been increased and units have been added to colorscale.*

*Fig 7b: we have labelled the first principal components.*

*Fig 11, 12, 13: we consider the figures are illustrative of the variability of the estimations considering the different sources of uncertainties, so we have kept them.*

**Reviewer 1: Minor and textual comments**

**ERA-Interim is known to have a serious dry bias in the interior ice sheet (doi.org/10.1175/2011JCLI4074.1).**

*Authors:*

*The authors thank reviewer 1 for pointing this out. To see the effect of the reanalysis, we had chosen to repeat the exercise with another reanalysis (ERA5), and included this the dataset (see section 3.1). Similar performances were obtained, even though some regional differences are present, driving at some places slightly different trends when applying to CESM2 runs (eg figure 12, bottom time series - DIR). This has been made clearer in the new version of the manuscript. (p.14, l. 237-240 "The choice of reanalysis used for training the analogs database has an effect on the downscaling obtained from CESM2 , as there still remain biases in reanalyses, e.g. a dry bias in the interior of the ice sheet (Bromwich et al., 2011). The choice of reanalysis influences the result, at equivalent training scores (Figure 12). This is one of the reasons for extending the database to ERA5 as training reanalysis instead of ERA-Interim.")*

**Reviewer 1: 850 hPa values for humidity, temperature are used, but this pressure level intersects with the surface of the ice sheet. How was this dealt with?**

*Authors:*

*As we have tested several levels (500, 700, 850 hPa) and different domain sizes, fields have been interpolated or extrapolated whenever necessary. (p.4, l.96-98: "Therefore, ERA-Interim fields of relative humidity at 850 , 700 and 500 hPa, geopotential height at 500 hPa, mean sea level pressure, sea ice extent, sea surface temperature and air temperature at 850 , 700 and 500 hPa were extracted from ECMWF archives and eventually extrapolated.") Finally, 700 hPa level for relative humidity was chosen, as the scores were better than 500 hPa, and similar to scores at 850 hPa, but without requiring interpolation. The choice was already mentioned in the method section (section 2.2), but it has been formulated more clearly in the revised manuscript. (p.6, l.149-151: "The analysis over the validation period (1991-2000) indicates that the optimal choice is reached with four fields: 1) geopotential height at 500 hPa, 2) surface pressure, 3) relative humidity at 700 hPa and 4) total precipitation.")*

**Reviewer 1: Abstract: the first few sentences do not really belong in an abstract, but rather in the introduction.**

*Authors:*

*Thank you very much for the suggestion. We have modified the abstract as follows.*

*We explore a methodology to statistically downscale snowfall – the primary driver of surface mass balance in Antarctica – from an ensemble of historical (1850-present day) simulations performed with an Earth System Model over the coastal region of Dronning Maud Land (East Antarctica). This approach consists in associating daily snowfall simulations from a polar-oriented Regional Atmospheric Climate Model at 5.5 km spatial resolution with specific weather patterns observed over 1979-2010 AD with the atmospheric reanalyses ERA-Interim and ERA5. This association is then used to generate the spatial distribution of snowfall for the period from 1850 to present day for an ensemble of ten members from the Earth System Model CESM2. The new dataset of daily and yearly snowfall accumulation based on this methodology is presented in this paper (MASS2ANT dataset, http://doi.org:10.5281/zenodo.4287517, Ghilain et al. (2021)). Based on a comparison with available ice cores and spatial reconstructions, our results show that the spatial-temporal*

*distribution of snowfall is improved in the downscaled dataset compared with the CESM2 simulations. This dataset thus provides information that may be useful in identifying the large-scale patterns associated with the local precipitation conditions and their changes over the past century.*

**Reviewer 1:**

**l. 1: over -> of**

**l. 9: Dronning Maud -> Dronning Maud Land**

**l. 18: the global -> global,**

**l. 31: reconstitutions?**

**l. 114: lead -> led**

**Table 1: please adjust the number of decimals of the coordinates**

*Authors:*

*Thank you  for noticing these errors, the text has been changed accordingly.*

**Reply to Reviewer 2's comments**

**Reviewer 2: The paper estimates annual precipitation rates over the coastal area of Dronning Maud Land. Using a downscaling from general circulation models of the atmosphere, a regional model and reanalysis data, the paper attempts to identify key atmospheric processes predicted back in time and which can be used to infer surface precipitation rates. The idea is intriguing and worthy of study. It is consistent with the theme of this journal.**

*Authors:*

*The authors thank the reviewer  for the careful reading of the manuscript and the comments.*

**Reviewer 2: The paper requires editorial revision and further analysis. Specifically, the paper repeatedly states that the results are an estimate of surface mass balance. In fact and as is also stated, the estimate is surface precipitation rates. Further reference to Lenaeerts et all 2019 and the magnitude of other contributions to SMB would be useful.**

*Authors:*

*Over Antarctica, SMB variability has been shown to be mostly driven by snowfall, with a contribution often one to two orders of magnitude larger than the other components (sublimation, melting, runoff, snow ablation by wind). Lenaerts et al (2019) has now been cited in the introduction. (p.2 l.26 & 44) This very strong link between precipitation and SMB has to be noted, because ice cores provide measurements of the SMB, and precipitation is a good proxy for the comparison with those measurements. However, we have carefully revised the manuscript to make a clear distinction when SMB can be mentioned, and when precipitation should be used instead.*

**Reviewer 2: Section 3.1 on the dataset should go at the end of the paper once a convincing argument about the utility of the data set has been made. To that point, the variation between model and ice core data seems to be about 25 % (Figure 8) and it needs to be made clear this is a meaningful result. Moreover and perhaps more importantly, there does not seem to be a strong correlation between the variations in the model data and the ice core data.  If there is a**

**correlation it should be quantified. I expected to read a conclusion about long term trends in precip rates as hinted at in the introduction. I did not find that nor do I think such a conclusion is possible based on inspection of Figure 13. More discussion on that point would be useful and would strengthen the paper whether a trend is discernible or not.**

*Authors:*

*We propose to maintain the information provided in Section 3.1 at the beginning of section 3, as in the submitted manuscript. It describes what information is accessible. We believe that this journal is precisely oriented toward the description of such a dataset and is thus at the core of the paper. The rest of the section illustrates the different parts of the accessible dataset that can be of interest for further analyses. The motivations (and utility) of the dataset has been re-stated at the end of the introduction. (p.3, l.67-72: "This dataset aims at providing an improved representation of detailed spatio-temporal patterns of snowfall over Dronning Maud Land. Firstly, this dataset allows for a more accurate comparison of long-term trends and temporal variability with local measurements (e.g., ice core snow accumulation records). Secondly, it enables a better understanding of the potential differences between estimates at small scale (such as the ones derived from ice cores) and the ones at the scale of the grid of a GCM. Finally, it makes it easier to associate synoptic meteorological conditions to snowfall events that determine the annual totals") Correlations with ice core data have been computed for each member of the downscaling, but have not been added to the manuscript, as we find that the exercise would not render any very interesting information to the reader. Indeed, the climate simulations were run in a 'historical' set-up with the same large-scale forcings (solar, atmospheric composition), but with initial conditions not necessarily reflecting the observed state, and without assimilating any observations. Therefore, temporal correlations with observations are not expected, nor is the downscaling expected to improve these correlations. Instead, the downscaling aims to better represent the natural variability. Multi-decadal trends have also been computed for each time series, and are discussed in the revision.*

**Reviewer 2: I made editorial comments in the text. My recommendation is to return the paper to the authors and to strongly encourage them to make major revisions before resubmitting. I believe they are on the right track to an interesting result but it needs a bit more work**

*Authors:*

*Thank you for the comments on the manuscript.*

- *As said, reference to SMB and precipitation have been carefully checked in the text.*

- *Inconsistencies in the introduction have been checked, eg ice shelf/ice sheet. (l.16-24)*

- *Snowfall is derived from CloudSat: SMB has been replaced by snowfall (l.90: 'where the precipitation was found to be consistent with satellites observations of CloudSat')*

- *We have rephrased l.108-115, so as to better express the first trials and why they failed, to justify the method used. ("Two methods have first been unsuccessfully tried: First, we tried to find a relation between the origin of moist air masses and the intensity of associated high precipitation amounts. However, the analysis of back-trajectories (on-line HYSPLIT model applied on NCEP/NCAR reanalysis (Stein et al. , 2015)) from strong snowfall events on Dronning Maud Land has not identified a strong relation. This result contrasts with the success obtained with ERA-Interim and ERA5 in identifying the atmospheric rivers responsible of such events (Gorodetskaya et al., 2020). Secondly, another approach based*

*on the Random Forest technique was tested, using as predictors several local atmospheric variables (air temperature, humidity, geopotential) or their large scale patterns. This method failed, probably due to the highly non-linear and non-systematic relations between large scale atmospheric patterns and snowfall intensity.")*

- *Zenodo is now explained: "Zenodo, an open-access repository operated by CERN".*

- *We have ensured that there is no confusion in the text between CESM member and downscaled member.*